# Evaluating situational judgment test use and diversity in admissions at a southern US medical school

Chelsea E. Gustafson[1,‡], Crystal J. Johnson[1,‡], Gary L. Beck Dallaghan[2]*, O'Rese J. Knight[3], Kimberly M. Malloy[4], Kimberley R. Nichols[5], Lisa Rahangdale[4]

**1** Medical Student, University of North Carolina School of Medicine, Chapel Hill, North Carolina, United States of America, **2** Department of Medical Education, University of Texas at Tyler School of Medicine, Tyler, Texas, United States of America, **3** Department of Otolaryngology, University of North Carolina School of Medicine, Chapel Hill, North Carolina, United States of America, **4** Department of Obstetrics-Gynecology, University of North Carolina School of Medicine, Chapel Hill, North Carolina, United States of America, **5** Department of Anesthesiology, University of North Carolina School of Medicine, Chapel Hill, North Carolina, United States of America

☯ These authors contributed equally to this work.
‡ CEG and CJJ are joint lead authors on this publication.
* gary.beckdallaghan@uthct.edu

**Data Availability Statement:** The data has been uploaded as a Supporting information file.

**Funding:** The author(s) received no specific funding for this work.

## Abstract

### Introduction

Situational judgment tests have been adopted by medical schools to assess decision-making and ethical characteristics of applicants. These tests are hypothesized to positively affect diversity in admissions by serving as a noncognitive metric of evaluation. The purpose of this study was to evaluate the performance of the Computer-based Assessment for Sampling Personal Characteristics (CASPer) scores in relation to admissions interview evaluations.

### Methods

This was a cohort study of applicants interviewing at a public school of medicine in the southeastern United States in 2018 and 2019. Applicants took the CASPer test prior to their interview day. In-person interviews consisted of a traditional interview and multiple-mini-interview (MMI) stations. Between subjects, analyses were used to compare scores from traditional interviews, MMIs, and CASPer across race, ethnicity, and gender.

### Results

1,237 applicants were interviewed (2018: n = 608; 2019: n = 629). Fifty-seven percent identified as female. Self-identified race/ethnicity included 758 White, 118 Black or African-American, 296 Asian, 20 Native American or Alaskan Native, 1 Native Hawaiian or Other Pacific Islander, and 44 No response; 87 applicants identified as Hispanic. Black or African-American, Native American or Alaskan Native, and Hispanic applicants had significantly lower CASPer scores than other applicants. Statistically significant differences in CASPer

**Competing interests:** The authors have declared that no competing interests exist.

percentiles were identified for gender and race; however, between subjects, comparisons were not significant.

## Conclusions

The CASPer test showed disparate scores across racial and ethnic groups in this cohort study and may not contribute to minimizing bias in medical school admissions.

## Introduction

Underrepresentation in US medical school from individuals identifying as Black or African-Americans, Hispanic, and American Indian or Alaska Native persists [1]. Underrepresented in medicine (UIM) physician role models lead to better outcomes for underrepresented minority patients [2]. Additionally, UIM students are nearly two times more likely to have plans to practice in underserved areas compared to their White and Asian counterparts [3]. There has been a steady decline in the numbers of African-American men entering medical school since 1978 [3–6]. Speculation about metrics such as grade point average (GPA) and Medical College Admission Test (MCAT) to select candidates for medical school may contribute to this decline [7].

Studies indicate that UIM individuals have lower GPA and MCAT scores [8,9]. Since GPA and MCAT scores are used to screen applicants for interviewing, overreliance on academic performance measures alone fails to capture noncognitive characteristics important for physicians [7]. Also, a meta-analysis reported the MCAT had minimal predictive value on future academic performance [10]. Structural racism and disparate educational opportunity play a role in MCAT performance of underrepresented applicants [11]. Consequently, overreliance on GPA and MCAT has the potential to lead to continued bias in admissions decisions.

Medical schools report balancing individual experience and attributes in conjunction with traditional metrics [12–14]. However, GPA and MCAT scores are heavily weighted in this screening process. Situational judgment tests add an alternative metric, assessing decision-making on predetermined scenarios [15]. Respondents judge the appropriateness of response choices by stating what they should or would do. Situational judgment tests in business demonstrated job performance predictive validity [16]. These tests have been used in hiring decisions and only recently have been used for medical school admissions [17].

The Computer-based Assessment for Sampling Personal Characteristics (CASPer) is a 12-section, online video-stem based situational judgment test of non-academic competencies. The test is structured to provide a scenario covering topics such as collaboration, communication, empathy, equity, ethics, motivation, problem solving, professionalism, resilience, and self-awareness. After each scenario, three follow up questions are presented that must be answered within five minutes using an open-ended response format.

Situational judgment tests are being used more by medical schools for holistic admissions processes. In fact, a review of the CASPer website indicates that 16 US medical schools and 6 osteopathic medical schools require students complete CASPer. The schools listed are geographically dispersed throughout the country.

The Office of Admissions for the University of North Carolina School of Medicine (UNCSOM) explored CASPer as an additional metric to use in admissions processes. Before integrating CASPer into the admissions process, we explored associations CASPer percentile scores had with our MMI and traditional interview scores. Our research questions included

the following: (1) What is the difference in CASPer percentile by gender and race/ethnicity? (2) What is the association between CASPer percentile with MMI and traditional interview scores?

## Material and methods

This cohort study included applicants to UNCSOM during the 2018 and 2019 admissions cycles. Demographic data extracted from the American Medical College Application Service (AMCAS) application included gender identification and race/ethnicity. The UNC Institutional Review Board reviewed this study and determined it met criteria for exempt status (IRB No. 18–3453) and met criteria under 45CFR46.116(f) to waive consent.

All applicants were required to take CASPer prior to their interview. Because we were still exploring the results, the UNCSOM did not formally use the CASPer percentile scores in admissions decisions. The test is taken by applicants on preset dates and times, requiring Internet access [15]. CASPer is comprised of up to 12 scenarios; each item comes with three questions students must provide a response. The test itself takes up to 120 minutes. CASPer is scored on a 9-point Likert scale (1 = unsatisfactory to 9 = superb). Each section is scored by a unique rater, thereby resulting in a score comprised of multiple raters. Responses are scored relative to other responses to the same scenario. Psychometric results of CASPer indicate overall test reliability (G = .72-.83) and inter-rater reliability (G = .82-.95) [18]. Students are informed about what quartile they achieved, but do not receive a specific percentile score. Only the medical school receives their percentile score.

On the interview day, applicants experience a 30-minute traditional interview with a faculty interviewer who had access to the applicant's AMCAS materials prior to the interview. Seven interpersonal and intrapersonal competencies were assessed by multiple mini-interviews (MMI) questions within two group stations (12–14 minutes) and two one-on-one stations (8 minutes each). A final station (8 minutes) allowed applicants to interact with a simulated patient as an introduction to our curriculum. Evaluators for the MMI stations were blinded to the applicant's AMCAS materials.

Applicant evaluations were scored 1–5 (low to high) for both traditional interview score and for each MMI station. Reliability across MMI stations resulted in Cronbach alpha = .66. Traditional interviewers assigned a score based on holistic review of the application and interview. MMI station interviewers were provided with a station-specific rubric for scoring, and an overall average MMI score was calculated.

To answer our first research question, CASPer percentile scores were compared by gender and UIM status. We generated a bivariate categorization of UIM using white and Asian as non-UIM and all others plus Hispanic as UIM. If significant differences were identified using the bivariate UIM categorization, analysis of variance (ANOVA) was conducted with racial groups of more than 100 individuals.

To answer the second research question, CASPer percentile, MMI and traditional interview scores were analyzed. Differences in scores were analyzed based on gender and UIM. Magnitude of factor differences was indicated by calculating Cohen's d, where Cohen's d = 0.2 is considered a "small" effect size, 0.5 "medium", and 0.8 "large" [19]. Regression analyses were conducted using MMI scores as the criterion variable and CASPer, gender, and UIM as predictors. All analyses were conducted using IBM SPSS v. 28 (Armonk, NY). Data is available as a Supporting Information file.

**Table 1. Demographic information for applicants.**

| | |
|---|---|
| Age | 24.02 (Range 20–41) |
| Gender | |
| Female | 699 (56.5%) |
| Male | 538 (43.5%) |
| Underrepresented in Medicine* | 213 (17.2%) |
| Race | |
| Asian | 296 (23.9%) |
| Black or African-American | 118 (9.5%) |
| Native Hawaiian or Other Pacific Islander | 1 (.1%) |
| Native American or Alaskan Native | 20 (1.6%) |
| White | 758 (61.3%) |
| Unanswered | 44 (3.6%) |
| Hispanic | 87 (7.0%) |

Percents are of the population sampled.

*Underrepresented in medicine was defined as Black or African-American, Native Hawaiian or Other Pacific Islander, Hispanic, and Native American or Alaskan Native.

## Results

There were 1,237 applicants interviewed during 2018 (n = 608) and 2019 (n = 629) admissions cycles. The average age of applicants was 24 years old. Fifty-seven percent identified as female. Demographic information is detailed in Table 1.

 To address our first research question, we used t-tests for bivariate comparisons. Table 2 summarizes the comparisons by gender, race, and ethnicity. In comparing gender, females scored higher for MMI (t = 3.77, p = .001, d = .21) and traditional interviews (t = 3.28, p = .001, d = .19). For CASPer percentile scores, UIM (t = -6.35, p = .001, d = .49) and Hispanic applicants (t = -3.28, p = .001, d = .38) scored lower. Given the significant differences of candidates identifying as UIM, we analyzed difference of candidates identifying as Asian, Black/African American, and White to obtain more specific results.

**Table 2. Comparisons of MMI Score, Traditional Interview Score and CASPer Percentile.**

| | Mean | Std. Dev. | Mean | Std. Dev. | t | p | Cohen's d |
|---|---|---|---|---|---|---|---|
| **Gender** | **Female (n = 698)** | | **Male (n = 537)** | | | | |
| MMI Average | 3.76 | .48 | 3.66 | .48 | 3.77 | .000 | .21 |
| Traditional Interview | 4.21 | .80 | 4.05 | .86 | 3.28 | .001 | .19 |
| CASPer Percentile | 61.05 | 26.76 | 58.50 | 27.97 | 1.59 | .111 | .09 |
| **UIM** | **Yes (n = 212)** | | **No (n = 1023)** | | | | |
| MMI Average | 3.74 | .50 | 3.71 | .48 | .84 | .402 | .07 |
| Traditional Interview | 4.11 | .83 | 4.15 | .83 | -.61 | .543 | .04 |
| CASPer Percentile | 48.92 | 27.51 | 62.18 | 26.73 | -6.35 | .000 | .49 |
| **Hispanic** | **Yes (n = 87)** | | **No (n = 1129)** | | | | |
| MMI Average | 3.63 | .50 | 3.72 | .48 | -1.68 | .093 | .19 |
| Traditional Interview | 4.01 | .83 | 4.15 | .83 | -1.51 | .131 | .17 |
| CASPer Percentile | 50.25 | 25.87 | 60.51 | 27.26 | -3.28 | .001 | .38 |

p significant < .05.

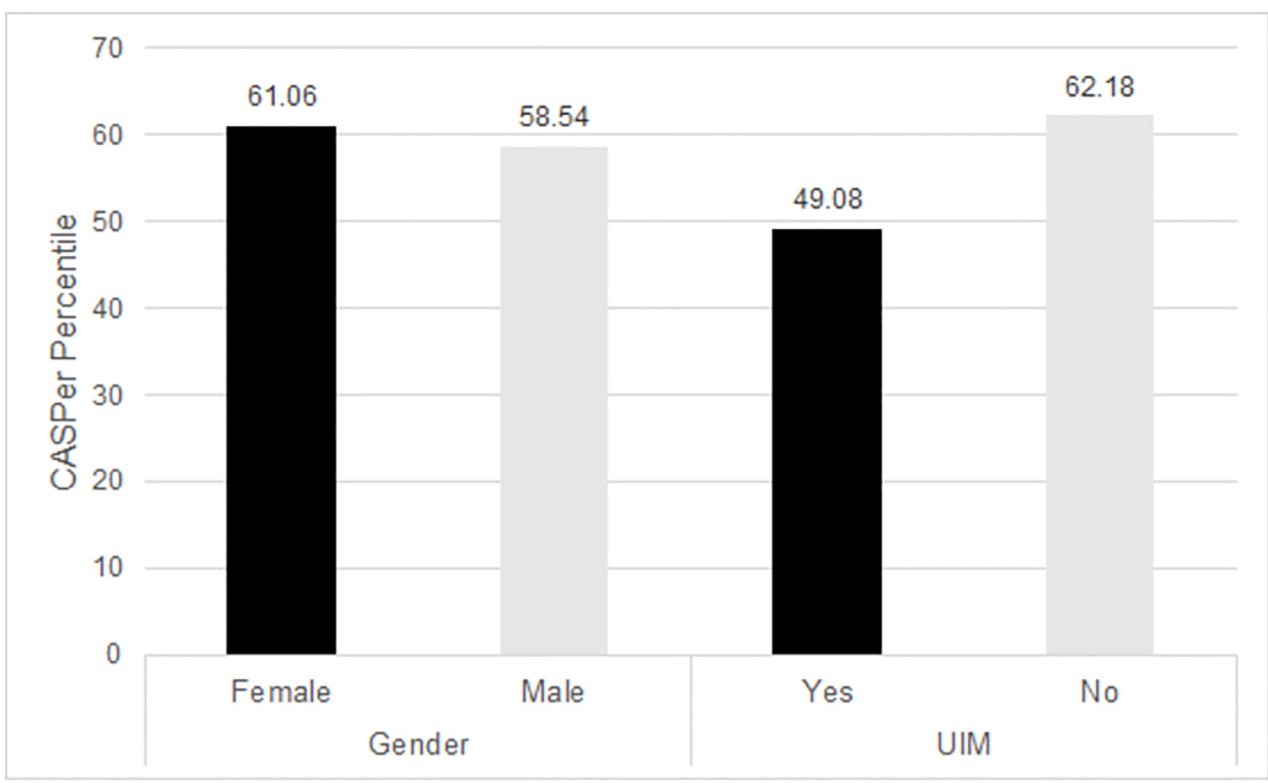

**Fig 1. Differences in CASPer Percentile by Gender, SES Indicator, and UIM Status.** A between subjects ANOVA indicated there were significant differences between CASPer percentile scores by each variable (Gender: $F_{2,1116}$ = 12.42, p < .001, $h^2$ = 0, UIM: $F_{2,1116}$ = 12.42, p < .001, $h^2$ = .02). Interactions between variables did not result in statistically significant differences.

The mean CASPer percentile score for Asian was 63.04 (n = 282, SD: 27.77, 95% CI:59.78–66.29), Black/African American was 47.86 (n = 112, SD: 28.74, 95% CI: 42.48–53.24), and White was 60.95 (n = 728, SD: 26.32, 95% CI: 59.04–62.87). The between groups model was significant for race ($F_{2,1116}$ = 12.42, p < .001, $h^2$ = .02). Post hoc analysis indicated Black/African American candidates had significantly lower CASPer percentile scores than Asian and White candidates (p < .001).

The between groups model for gender was also significant ($F_{2,1116}$ = 12.42, p < .001, $h^2$ = 0) for females scoring higher than males. The interaction of race and gender was not statistically significant. (Fig 1).

To address question two, a linear regression model explored an association with MMI score using CASPer percentile, gender, and race. All three predictors were significantly associated with MMI score (Constant: 3.68; CASPer: β = .16, t = 5.55, p < .001; Gender: β = -.10, t = -3.50, p < .001; Race: β = .07, t = 2.311, p = .02). However, these variables identified a weak correlation (r = .20), explaining approximately 4 percent of the variance in MMI scores.

## Discussion

Situational judgment tests contribute noncognitive data to consider as part of the medical school admissions process [20]. Based on our pool of candidates, CASPer scores varied significantly, with lower scores seen in those identifying as Black or African-American, American Indian or Alaskan Native, Native Hawaiian or Other Pacific Islander, or Hispanic ethnicity.

Our findings different from the New York Medical College School of Medicine study [15], which may be a reflection of the differences in the candidate pools. Specifically, in their comparison of White and African-American applicants, the White applicants scored higher on CASPer, but not significantly. As part of their study they used these results to simulate potential for an interview offer for medical school. With the inclusion of CASPer in the simulation, results suggested an increase in African-Americans being invited to interview.

Our results suggest UIM students may be further disadvantaged if CASPer was weighted in applicant screening. The UNCSOM candidate pool is from the southeastern US; perhaps social and cultural differences played a role in individual performance. Previous studies note differences based on race and ethnicity [21] as a result of response instructions for the test. This may be the result of the scoring process as well, since raters compare scenario responses as they grade. The CASPer test was also developed in Canada [22,23], but little research has been published using the instrument in the United States. Our results should be an example to other schools to regularly analyze results of instruments being used to ensure they are not inadvertently disadvantaging a particular population.

There is evidence of increased use of CASPer in medical school admissions. Additionally, a scan of the CASPer website indicates 22 US medical schools and osteopathic schools use the exam. Although medical schools are using the exam, there continues to be little agreement on what is actually being measured or how to integrate the information effectively in holistic review [24]. Altus Assessments has recommended using quartile performance to rate candidates [25], suggesting that acceptable scores range from the 33rd to 75th percentile. However, with such a broad range of scores one must question what these results actually mean and how do they pertain to future medical student performance.

When CASPer was analyzed by gender, there was not a significant difference in how females performed compared to males. These findings are consistent with work by Whetzel and colleagues [21]. However, females did significantly better on MMIs and the traditional interview, consistent with other studies demonstrating stronger female communication ratings [26,27]. Yet given a study demonstrating a correlation of situational judgment test items to interpersonal communication skills [24], one would expect females to have outperformed males on CASPer.

Based on two years of data from UNCSOM, MMI and traditional scores did not show differences across racial groups [26]. Since UNCSOM was piloting CASPer, the results of the test were not part of the formal deliberations by the admissions committee. If we continue using CASPer, applying methods outlined by Aguinis and Smith [28] may clarify an appropriate cut score for CASPer. They calculated relationships between desired performance levels, expected adverse impact, and probabilities of false positives and false negatives to determine a cut score. Alternatively, weighting interview metrics using a Pareto optimization approach could allow for institutional flexibility in predictive measures as well as diversity goals [29].

This study was conducted at a single institution in the southeastern US, presenting a limitation. Future studies should be conducted with other medical schools using CASPer to explore whether differences appear. As with any assessment, requiring applicants to complete CASPer may have contributed to performance on the examination. This is a question that should be further explored in future research.

## Conclusions

Using a variety of methods in the admissions process appears to be the best approach to ensure diversity and inclusion in medical student classes. Overreliance on MCAT scores or GPAs inadvertently disadvantages underrepresented minorities. Although well intended, the CASPer

was not found to level the playing field as we thought it might. Based on these data, using a combined approach of rating academic performance, preparation for medical school, life experience, interviews, and MMIs along with appropriately trained interviewers and committee members may be sufficient for holistic admissions that favors a diverse medical school class.

## Supporting information

**S1 File. This is the S1 File plos1CASPerAnalysis.**
(XLSX)

## Author Contributions

**Conceptualization:** Chelsea E. Gustafson, Crystal J. Johnson, Lisa Rahangdale.

**Data curation:** Gary L. Beck Dallaghan, Lisa Rahangdale.

**Formal analysis:** Chelsea E. Gustafson, Gary L. Beck Dallaghan, O'Rese J. Knight, Kimberley R. Nichols, Lisa Rahangdale.

**Investigation:** Crystal J. Johnson.

**Methodology:** Gary L. Beck Dallaghan, Lisa Rahangdale.

**Visualization:** Gary L. Beck Dallaghan, Kimberly M. Malloy, Kimberley R. Nichols.

**Writing – original draft:** Chelsea E. Gustafson, Crystal J. Johnson, Gary L. Beck Dallaghan, O'Rese J. Knight, Kimberly M. Malloy, Kimberley R. Nichols, Lisa Rahangdale.

**Writing – review & editing:** Chelsea E. Gustafson, Crystal J. Johnson, Gary L. Beck Dallaghan, O'Rese J. Knight, Kimberly M. Malloy, Kimberley R. Nichols, Lisa Rahangdale.

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
