## [Decision Letter · Decision Letter 0]

1 Sep 2022

PONE-D-22-17727Evaluating situational judgment test use and diversity in admissions at a southern US medical schoolPLOS ONE

Dear Dr. Beck Dallaghan,

Thank you for submitting your manuscript to PLOS ONE. After careful consideration, we feel that it has merit but does not fully meet PLOS ONE’s publication criteria as it currently stands. Therefore, we invite you to submit a revised version of the manuscript that addresses the points raised during the review process.

Please submit your revised manuscript before Oct 16 2022 11:59PM. If you will need more time than this to complete your revisions, please reply to this message or contact the journal office at plosone@plos.org. Please include the following items when submitting your revised manuscript:A rebuttal letter that responds to each point raised by the academic editor and reviewer(s). You should upload this letter as a separate file labeled 'Response to Reviewers'.A marked-up copy of your manuscript that highlights changes made to the original version. You should upload this as a separate file labeled 'Revised Manuscript with Track Changes'.An unmarked version of your revised paper without tracked changes. You should upload this as a separate file labeled 'Manuscript'.

We look forward to receiving your revised manuscript.

Kind regards,

Yaser Mohammed Al-Worafi

Academic Editor

PLOS ONE

2. Please provide additional details regarding participant consent. If the need for consent was waived by the ethics committee, please include this information.

Reviewers' comments:

Reviewer's Responses to Questions

**Comments to the Author**

1. Is the manuscript technically sound, and do the data support the conclusions?

Reviewer #1: Yes

2. Has the statistical analysis been performed appropriately and rigorously? 

Reviewer #1: Yes

3. Have the authors made all data underlying the findings in their manuscript fully available?

Reviewer #1: Yes

4. Is the manuscript presented in an intelligible fashion and written in standard English?

Reviewer #1: Yes

5. Review Comments to the Author

Reviewer #1: This is a fairly straightforward assessment of the CASPer test and whether this is a non-biased metric for minimizing bias in admissions for American medical schools. Results are important as data suggests that Black or African-American, Native American or Alaskan Native, and Hispanic applicants all had lower scores than other groups of applicants suggesting that this test may not be an accurate assessment of reducing bias in admissions. A few items should be added for clarity to the report.

1. Clarity on how this new data fits into the wholistic review of applicants is needed in the discussion. Expansion of this point is needed from what is currently in the report.

2. Is there any bias in forcing applicants to simply take the CASPer test that could have resulted in lower scores?

3. More information on the specifics of the CASper test could be added to the methods.

6. PLOS authors have the option to publish the peer review history of their article (what does this mean?). If published, this will include your full peer review and any attached files.

Reviewer #1: No

---

## [Author Response · Author response to Decision Letter 0]

15 Sep 2022

PONE-D-22-17727: Evaluating situational judgment test use and diversity in admissions at a southern US medical school

Response to Reviewers

Thank you for the opportunity to revise and resubmit our manuscript. We have taken all of the comments into consideration and have made revisions accordingly. Please see our detailed responses below for each one.

========

>>We have made revisions to the format of the manuscript based on the template guidelines. We have also renamed the figure and supplemental file names per your instructions.

2. Please provide additional details regarding participant consent. If the need for consent was waived by the ethics committee, please include this information.

>>We have revised the manuscript to indicate that consent was waived by the ethics committee in the Methods section of the manuscript. Their review of the project was deemed exempt and that consent was not required.

>>A blinded supplemental file has been uploaded with our data included. In order to ensure that no identifiers were present, names, identification numbers, and other demographic variables were removed.

>>The ethics statement is in the Methods section. We have removed it from other spots in the manuscript.

Reviewers' comments:

Reviewer's Responses to Questions

Comments to the Author

Reviewer #1: This is a fairly straightforward assessment of the CASPer test and whether this is a non-biased metric for minimizing bias in admissions for American medical schools. Results are important as data suggests that Black or African-American, Native American or Alaskan Native, and Hispanic applicants all had lower scores than other groups of applicants suggesting that this test may not be an accurate assessment of reducing bias in admissions. A few items should be added for clarity to the report.

1. Clarity on how this new data fits into the wholistic review of applicants is needed in the discussion. Expansion of this point is needed from what is currently in the report.

>>We did mention in the Methods that CASPer was not used as a formal part of the admissions process due to this being a pilot. We have also included some additional language in the Discussion section to that effect as well.

2. Is there any bias in forcing applicants to simply take the CASPer test that could have resulted in lower scores?

>>This is a good question and one that we cannot answer with the data we collected. However, we have incorporated a statement related to this as a potential limitation of the study and worthy of future study.

3. More information on the specifics of the CASper test could be added to the methods.

>>Additional information about the CASPer is included in the Methods.

---

## [Decision Letter · Decision Letter 1]

21 Nov 2022

PONE-D-22-17727R1Evaluating situational judgment test use and diversity in admissions at a southern US medical schoolPLOS ONE

Dear authors,

Thank you for submitting your manuscript to PLOS ONE. After careful consideration, we feel that it has merit but does not fully meet PLOS ONE’s publication criteria as it currently stands. Therefore, we invite you to submit a revised version of the manuscript that addresses the points raised during the review process.

We look forward to receiving your revised manuscript.

Kind regards,

Yaser Mohammed Al-Worafi

Academic Editor

PLOS ONE

Reviewers' comments:

Reviewer's Responses to Questions

**Comments to the Author**

1. If the authors have adequately addressed your comments raised in a previous round of review and you feel that this manuscript is now acceptable for publication, you may indicate that here to bypass the “Comments to the Author” section, enter your conflict of interest statement in the “Confidential to Editor” section, and submit your "Accept" recommendation.

Reviewer #2: All comments have been addressed

Reviewer #3: All comments have been addressed

Reviewer #4: All comments have been addressed

2. Is the manuscript technically sound, and do the data support the conclusions?

Reviewer #2: Yes

Reviewer #3: Yes

Reviewer #4: Partly

3. Has the statistical analysis been performed appropriately and rigorously? 

Reviewer #2: Yes

Reviewer #3: Yes

Reviewer #4: I Don't Know

4. Have the authors made all data underlying the findings in their manuscript fully available?

Reviewer #2: Yes

Reviewer #3: Yes

Reviewer #4: Yes

5. Is the manuscript presented in an intelligible fashion and written in standard English?

Reviewer #2: Yes

Reviewer #3: Yes

Reviewer #4: Yes

6. Review Comments to the Author

Reviewer #2: The authors have appropriately addressed all reviewers’ comments. The manuscript is ready for publication.

Reviewer #3: Dear authors,

Thank you for addressing the previous reviews. In addition to the current revision, would it be possible to provide further information regarding CASper instrument? You mentioned that the reliability of the item is 0.66. It will be useful for the readers if you can describe further what topics/subjects covered by CASper, as well as provide the example of the items. If this has been reported elsewhere, of course the authors may also cite the report. I suggest this to be elaborated in the method section, hence the authors might be able to refer this in the discussion section. This study highlights that the CASper cannot be used as the selection instrument, despite the initial intention. I wonder what's the implication of this, whether the authors would be able to do further actions on the items or not?

Reviewer #4: This manuscript reads more to me like a research letter. The authors tried an intervention to specifically broaden selection criteria that might holistically reduce racial/ethnic bias, but it’s unclear that there was any rationale that this tool might do that. If anything the authors could reframe their original hypothesis and manuscript in this manner, i.e. saying that there is no evidence to suggest it would however they sought to explore whether it would and it did not seem to. And perhaps even does the opposite in perpetuating biases.

Intro:

Since CASPER is new to this audience, could use a little more background in the Intro as to how it was validated. Has it previously shown bias or better been shown to not perpetuate bias and inequities in diverse samples?

Was NHPI analyzed with Asian or with UIM. It should be with UIM. It’s unclear in the Methods and Results as it seems to be state both ways.

I’d suggest formal Stats review.

Do the CASPER results differ by race when controlling for other quantitative metrics like GPA or MCAT score?

Discussion:

The Discussion in general is way too short and superficial / skimpy.

What did the NYU study show? Should clarify in more detail in the Discussion itself.

Paragraph 2 gets to the crux of my initial question. If CASPER was introduced as a tool to mitigate bias, increase diversity, and enhance holistic review, where is the evidence that it actually does that? Especially if there are subjective components to the scoring which would be at risk of implicit and explicit biases.

“applying Methods from”. This statement should be similarly elaborated. What unique recommendations did they have?

7. PLOS authors have the option to publish the peer review history of their article (what does this mean?). If published, this will include your full peer review and any attached files.

Reviewer #2: No

Reviewer #3: No

Reviewer #4: No

---

## [Author Response · Author response to Decision Letter 1]

19 Dec 2022

Reviewer #2: The authors have appropriately addressed all reviewers’ comments. The manuscript is ready for publication.

>>Thank you.

Reviewer #3: Dear authors,

Thank you for addressing the previous reviews. In addition to the current revision, would it be possible to provide further information regarding CASper instrument? You mentioned that the reliability of the item is 0.66. It will be useful for the readers if you can describe further what topics/subjects covered by CASper, as well as provide the example of the items. If this has been reported elsewhere, of course the authors may also cite the report. I suggest this to be elaborated in the method section, hence the authors might be able to refer this in the discussion section. This study highlights that the CASper cannot be used as the selection instrument, despite the initial intention. I wonder what's the implication of this, whether the authors would be able to do further actions on the items or not?

>>Thank you for this comment. We have added additional information in the introduction as well as Methods related to CASPer. Since it is a proprietary product, we would not have any ability to influence the items in the instrument. We did comment more on the fact that this instrument could be considered part of holistic review of materials, but that institutions need to do a good job of continuous quality improvement to ensure that they understand the scores for their context.

Reviewer #4: This manuscript reads more to me like a research letter. The authors tried an intervention to specifically broaden selection criteria that might holistically reduce racial/ethnic bias, but it’s unclear that there was any rationale that this tool might do that. If anything the authors could reframe their original hypothesis and manuscript in this manner, i.e. saying that there is no evidence to suggest it would however they sought to explore whether it would and it did not seem to. And perhaps even does the opposite in perpetuating biases.

>>We appreciate your comments. Unfortunately, little has been published about the fact that many medical schools are using CASPer now and consider it to be unbiased. NY Medical College reports these data, but others have not. Anecdotally, we thought it would provide a more unbiased measure based on the data provided by CASPer. That is the reason we wanted to explore the results and thus asked the questions we did.

Intro:

Since CASPER is new to this audience, could use a little more background in the Intro as to how it was validated. Has it previously shown bias or better been shown to not perpetuate bias and inequities in diverse samples?

>>Thank you for this comment. Where CASPer is concerned, no such reports to our knowledge exist. There are other types of situational judgment tests that have shown bias toward ethnic groups, but not in medicine. We discuss that in the Discussion. 

Was NHPI analyzed with Asian or with UIM. It should be with UIM. It’s unclear in the Methods and Results as it seems to be state both ways.

>>NHPI was not part of “Asian” and this was corrected in the manuscript.

I’d suggest formal Stats review.

>>We have an experienced statistician on our team who reviewed materials again.

Do the CASPER results differ by race when controlling for other quantitative metrics like GPA or MCAT score?

>>We did not incorporate GPA and MCAT in this analysis. The reason being is that those are primarily considered knowledge-based outcomes. CASPer is meant for non-cognitive abilities and therefore using interview evaluation metrics was more appropriate for an analysis.

Discussion:

The Discussion in general is way too short and superficial / skimpy.

>>We have added to the Discussion.

What did the NYU study show? Should clarify in more detail in the Discussion itself.

>>We have added to the Discussion.

Paragraph 2 gets to the crux of my initial question. If CASPER was introduced as a tool to mitigate bias, increase diversity, and enhance holistic review, where is the evidence that it actually does that? Especially if there are subjective components to the scoring which would be at risk of implicit and explicit biases.

>>We’ve further discussed some of this as part of the expanded discussion. As we note throughout, situational judgment tests are being used in business and medical school admissions as a means of mitigating bias. However, few people have reported findings like ours. Why? We don’t know. But pointing out how different these results are based on UIM status is worth reporting because there may be issues with the grading that CASPer does. As a private business, we have no control over their process.

“applying Methods from”. This statement should be similarly elaborated. What unique recommendations did they have?

>>We have revised this.

---

## [Editor Report · Decision Letter 2]

22 Dec 2022

Evaluating situational judgment test use and diversity in admissions at a southern US medical school

PONE-D-22-17727R2

Dear colleagues, 

We’re pleased to inform you that your manuscript has been judged scientifically suitable for publication and will be formally accepted for publication once it meets all outstanding technical requirements.

Kind regards,

Yaser Mohammed Al-Worafi

Academic Editor

PLOS ONE

---

## [Editor Report · Acceptance letter]

29 Dec 2022

PONE-D-22-17727R2 

Evaluating situational judgment test use and diversity in admissions at a southern US medical school 

Dear Dr. Beck Dallaghan:

I'm pleased to inform you that your manuscript has been deemed suitable for publication in PLOS ONE. Congratulations! Your manuscript is now with our production department. 

Kind regards, 

on behalf of

Professor Yaser Mohammed Al-Worafi 

Academic Editor

PLOS ONE